

# Taxonomic annotation errors incorrectly assign the family Pseudoalteromonadaceae to the order Vibrionales in Greengenes: implications for microbial community assessments

Keri Ann Lydon* and Erin K. Lipp*

Department of Environmental Health Science, University of Georgia, Athens, GA, USA
* These authors contributed equally to this work.

## ABSTRACT

Next-generation sequencing has provided powerful tools to conduct microbial ecology studies. Analysis of community composition relies on annotated databases of curated sequences to provide taxonomic assignments; however, these databases occasionally have errors with implications for downstream analyses. Systemic taxonomic errors were discovered in Greengenes database (v13_5 and 13_8) related to orders Vibrionales and Alteromonadales. These orders have family level annotations that were erroneous at least one taxonomic level, e.g., 100% of sequences assigned to the Pseudoalteromonadaceae family were placed improperly in Vibrionales (rather than Alteromonadales) and >20% of these sequences were indeed *Vibrio* spp. but were improperly assigned to the Pseudoalteromonadaceae family (rather than to Vibrionaceae). Use of this database is common; we identified 68 peer-reviewed papers since 2013 that likely included erroneous annotations specifically associated with Vibrionales and Pseudoalteromonadaceae, with 20 explicitly stating the incorrect taxonomy. Erroneous assignments using these specific versions of Greengenes can lead to incorrect conclusions, especially in marine systems where these taxa are commonly encountered as conditionally rare organisms and potential pathogens.

Corresponding authors
Keri Ann Lydon,
keri.lydon@gmail.com
Erin K. Lipp, elipp@uga.edu

## INTRODUCTION

Analysis of 16S rRNA gene sequences has dramatically changed the way microbiologists understand the ecology of whole bacterial communities in an ecosystem. We are now able to sequence millions of reads of this gene in mixed samples to understand changes and dynamics in microbial composition. To analyze these data, sequence reads are compared against a curated ribosomal sequence database with known taxonomic identities. Frequently used databases include Greengenes (*DeSantis et al., 2006*; *McDonald et al., 2012*; http://greengenes.secondgenome.com), SILVA (*Pruesse et al., 2007*),

the ribosomal database project (RDP) (*Cole et al., 2014*), or NCBI (*Federhen, 2012*). Greengenes is highly cited and was first introduced in 2006 for assigning taxonomies to Archaea and Bacteria using an automated *de novo* tree-based approach (*DeSantis et al., 2006*; *McDonald et al., 2012*). One of the reasons for its popularity is its ease of use given that it has been assimilated into 16S rRNA gene sequence analysis pipelines such as QIIME (*Caporaso et al., 2010*). Although Greengenes is one of the smallest databases, it has been suggested as the preferred database for classification of taxonomy because of its capacity to assign taxonomy to great depth (e.g., species level identification) (*Werner et al., 2012*).

In a recent analysis of 16S rRNA gene sequences using QIIME version 1.9.1 (*Caporaso et al., 2010*) in our laboratory, we observed that the Greengenes taxonomy for the orders Vibrionales and Alteromonadales appeared to have errors when using version 13_8. Further investigation of Greengenes versions revealed that these errors first appeared in version 13_5, but were not present in earlier versions (accessed March 26, 2018; ftp://ftp.microbio.me/greengenes_release/).

The taxonomic orders Vibrionales and Alteromonadales include heterotrophic Gram-negative bacteria within the class Gammaproteobacteria that are important members in a number of environments, especially in marine systems. The genus *Vibrio* in Vibrionales includes opportunistic, ubiquitous, and conditionally rare taxa that are studied for their contributions to nutrient cycling in the marine environment (*Thompson & Polz, 2006*; *Takemura, Chien & Polz, 2014*; *Vezzulli et al., 2016*), and more notably pathogenicity in both invertebrates and vertebrates (*Austin & Zhang, 2006*). Furthermore, the family Vibrionaceae includes several common human pathogens (*Newton et al., 2012*; *U.S. Center for Disease Control and Prevention (CDC), 2016*). Pseudoalteromonadaceae, which belong to the Alteromonadales order, are also opportunistic and conditionally rare in the marine environment, where they have been associated with disease (*Choudhury et al., 2015*; *Beurmann et al., 2017*) and as biofilm forming bacteria associated with the surfaces of eukaryotic organisms (*Davis et al., 1989*; *Holmström & Kjelleberg, 1999*). They occupy a similar niche as Vibrionaceae, yet *Pseudoalteromonas* spp. (within the family Pseudoalteromonadaceae) are known antagonists of *Vibrio* spp. (*Morya, Choi & Kim, 2014*; *Richards et al., 2017*).

*Edgar (2018a)* published a non-peer reviewed pre-print (March 25, 2018) that found the annotation error rate in Greengenes version 13_5 to be approximately 15%, including mismatches for the families Vibrionaceae and Pseudoalteromonadaceae between SILVA version 128 and Greengenes 13_5 (amended to a 17% error rate in a revised version (*Edgar, 2018b*)). Although the analyses reported by *Edgar (2018b)* suggest broader issues with taxonomic mismatches (also recognized by *DeSantis et al., 2006*, with the original release of Greengenes) among others (*Beiko, 2016*; *Kozlov et al., 2016*; *Balvočiūtė & Huson, 2017*), we specifically assessed the extent of the misclassification associated with order Vibrionales and family Pseudoalteromonadaceae, given the importance of these taxa in marine systems. Furthermore, we highlight that these errors do not appear to be due to real differences in inferred taxonomy between databases but are rather an example of a specific assignment error in Greengenes curation.

## METHODS

### Taxonomic comparison of Pseudoalteromonadaceae across curated databases

We searched for the bacterial family Pseudoalteromonadaceae within the representative sequences and taxonomy for Greengenes version 13_8 provided within QIIME version 1.9.1 and found that 100% of Pseudoalteromonadaceae sequences ($n = 164$) were misclassified within the Vibrionales order when they properly belong to the Alteromonadales order (File S1). This was also observed in an earlier version of Greengenes (13_5). Subsequently, we checked taxonomies against NCBI (16 rRNA gene database), RDP (Type Strains), and SILVA (ref NR) using the web interfaces of BLAST (*Morgulis et al., 2008*; https://blast.ncbi.nlm.nih.gov), SeqMatch (*Cole et al., 2014*; http://rdp.cme.msu.edu/seqmatch/seqmatch_intro.jsp), and SINA (*Pruesse, Peplies & Glöckner, 2012*; https://www.arb-silva.de/aligner/), respectively, for all Pseudoalteromonadaceae sequences from the Greengenes representative sequences in version 13_8 to determine if there were additional mismatches at other levels of taxonomy.

### Phylogenetic tree of Pseudoalteromonadaceae in Greengenes

The 16S rRNA gene representative sequences assigned to Pseudoalteromonadaceae in Greengenes ($n = 164$) were used to create a phylogenetic tree in Unipro UGENE v.1.26.3 (*Okonechnikov et al., 2012*). Reads were first aligned using MUSCLE (*Edgar, 2004*), before subsequent tree building using PhyML maximum likelihood with aLRT SH-like branch support (*Guindon et al., 2010*). Results were visualized in iTOL (version 4.0.3; *Letunic & Bork, 2007*), including the addition of color strips which indicate the assigned taxonomy results generated in this study from NCBI, RDP, SILVA, and Greengenes queries.

### Literature search for taxonomic errors

We attempted to examine how widespread such a taxonomic annotation error might have been propagated in the microbial ecology literature. We conducted a systematic literature search within Google Scholar using the terms "Greengenes + Vibrionales + Pseudoalteromonadaceae" or "Greengenes + Vibrionales" for the years 2013–2018 (Greengenes 13_5 and 13_8 were released in 2013), including only published peer-reviewed papers accessible in English and which used a next-generation amplicon sequencing approach (i.e., excluding analyses with PhyloChip, DGGE, or clone libraries). The search was completed between March 23 and 28, 2018.

## RESULTS

All of the 164 sequences identified as Pseudoalteromonadaceae in the Greengenes database were incorrectly assigned to the Vibrionales order, rather than the Alteromonadales order (the accepted next level of taxonomic rank for this family (*Ivanova, Ng & Webb, 2014*)). Of these 164 representative sequences in Greengenes, 115, 118, and 117 were confirmed by SILVA, RDP, and NCBI, respectively, as Pseudoalteromonadaceae sequences and were assigned (correctly) to the order Alteromonadales. Furthermore, RDP and NCBI identified 45, and SILVA 43, of the 164 sequences as *Vibrio* spp. that were incorrectly

assigned to the Pseudoalteromonadaceae family in Greengenes; therefore, because Greengenes assigned all members of the Pseudoalteromonadaceae family to the Vibrionales order, these 43–46 sequences were only incorrect at the family level (Fig. 1). These sequences mismatched by family represented 23 *Vibrio* spp. and included common species such as *V. alginolyticus, V. cholerae,* and *V. harveyi* (File S1). This means that while these sequences did belong in the order Vibrionales, if the family level was chosen for downstream analysis, they would have been analyzed incorrectly as Pseudoalteromonadaceae. There were several sequences for which we were unable to confirm taxonomic identity using NCBI, SILVA, or RDP or were attributed to the bacterial families Shewanellaceae, Colwelliaceae, Oceanospirillaceae within the order Alteromonadales (Fig. 1).

We found 85 published papers that met our search criteria in journals such as *PLOS ONE, Frontiers in Microbiology, ISME Journal,* among others that used either 13_5 or 13_8 versions of Greengenes to assign their taxonomy (Table 1). Of these articles, 20 reported results clearly stating the incorrect taxonomic assignments or listing the mismatched taxonomy in a table (in text or in supplemental material). An additional 35 papers were presumed to have misclassification errors based on the database version reported. Only 17 of the evaluated articles reported either the correct taxonomic assignment directly or were presumed to have the correct assignment based on the use of earlier Greengenes versions (or the explicitly stated use of other databases for assignment), which did not have the mismatch error. The remaining 13 papers did not report sufficient detail on the database or bioinformatics pipeline used to determine if the mismatch error was likely. In all, 55 of the 85 papers (~64.7%) since 2013 likely reported relative abundance levels of Vibrionales or Alteromonadales that were incorrect due to improper taxonomic assignments.

## DISCUSSION AND CONCLUSIONS

Results presented here demonstrate the extent of taxonomic annotation errors associated with the family Pseudoalteromonadaceae present in the Greengenes database versions 13_8 and 13_5 for the order Vibrionales. Although the Gammaproteobacteria are known to include a number of polyphyletic groups, there is a clear distinction between the Vibrionales and Alteromonadales (*Williams et al., 2010*). Given that this assignment of Pseudoalteromonadaceae to Vibrionales only occurred in the Greengenes databases (13_5 and 13_8), this suggests an issue of curation rather than an underlying change in real taxonomic assignment. The Vibrionales and Alteromonadales are often associated with similar niches in marine systems; however, correct taxonomic classification is imperative, especially when attribution is important, such as the identification of potential pathogens.

Issues with taxonomic assignments among databases have been the subject of previous and on-going research (*Kozlov et al., 2016*; *Edgar, 2018b*), given its importance in studies where identification is needed. Results with erroneous attribution can alter our understanding or interpretation of the ecology of those microbial assemblages. For example, given that 27% of *Vibrio* spp. sequences in Greengenes were incorrectly

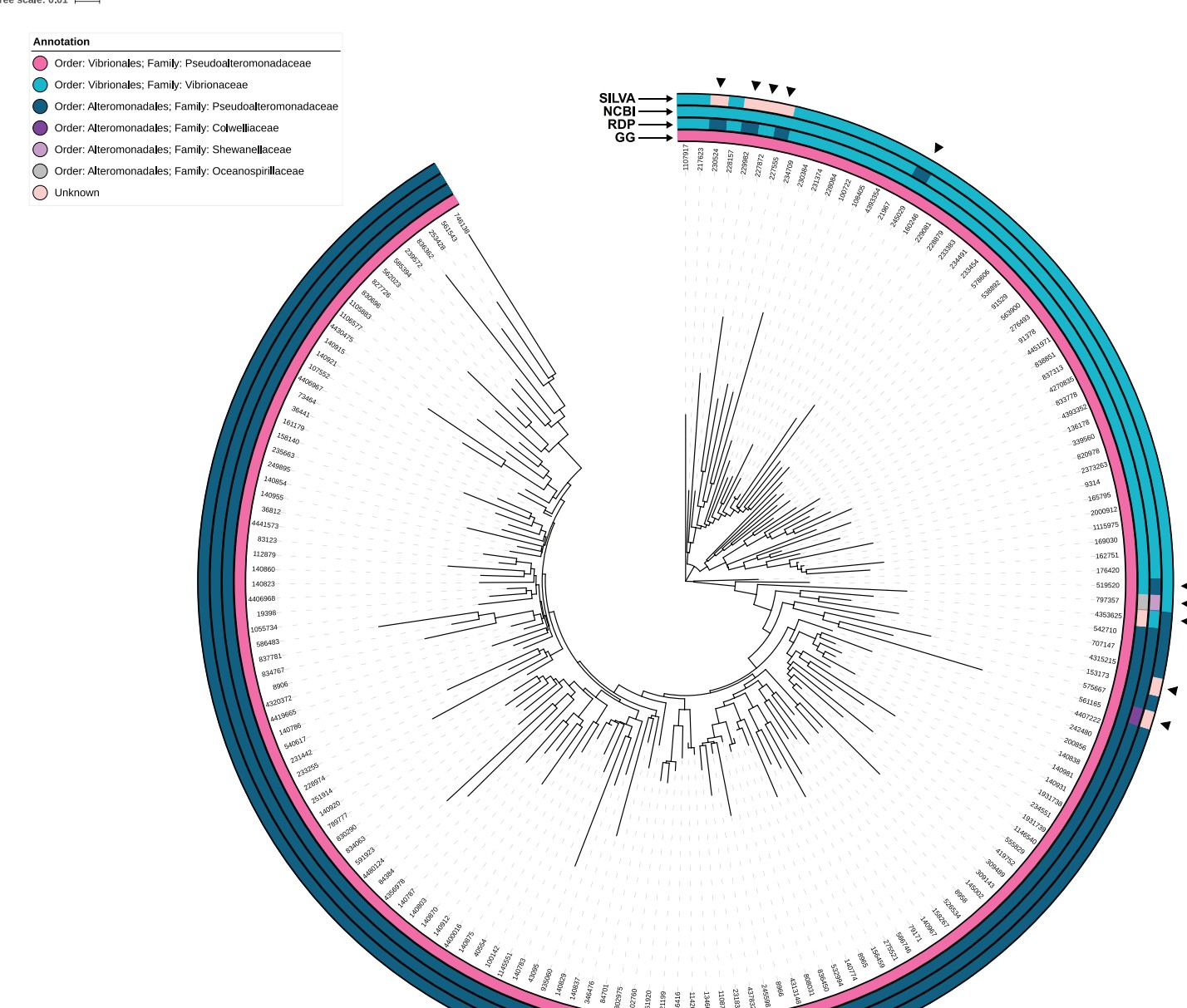

**Figure 1** **Phylogenetic tree of Pseudoalteromonadaceae representative sequences in the Greengenes database ($n$ = 164).** Each branch is labeled by the assigned ID in the representative sequences within Greengenes. Color strips indicated the assigned taxonomy (Order and Family) from different curated databases: Greengenes (GG), SILVA, RDP, and NCBI. The accepted Order for the family Pseudoalteromonadaceae is Alteromonadales (*Ivanova, Ng & Webb, 2014*) not Vibrionales, as featured in all Greengenes taxonomic assignments.

attributed to Pseudoalteromonadaceae instead of Vibrionaceae, we could be making inferences on artificially low relative abundances at the family level. Likewise, at the order level Pseudoalteromonadaceae sequences could artificially inflate the relative abundance of Vibrionales because of these assignment errors.

**Table 1 Incorrect taxonomic annotation of Pseudoalteromonadaceae in Vibrionales as published in peer reviewed literature.**

| Search term queries | Greengenes + Vibrionales + Pseudoalteromonadaceae | Greengenes + Vibrionales[a] | Total |
|---|---|---|---|
| All papers fitting search criteria | 22 | 63 | 85 |
| Papers confirmed using Greengenes versions 13_5 or 13_8 with known taxonomy errors | 14 | 41 | 55 |
| Papers explicitly stating taxonomic mismatch (in text or supplemental material)[b] | 10 | 10 | 20 |
| Papers stating correct assignment or using earlier version of Greengenes | 7 | 10 | 17 |
| Papers using Greengenes but with no information on database version. | 1 | 12 | 13 |

Notes:

Search was conducted between March 23 and 28, 2018 using Google Scholar. Search results included in the analysis were peer-reviewed papers published between 2013 and 2018 that were accessible in English and used a 16S rRNA gene next-generation sequencing approach.

[a] Counts do not include papers appearing in the "Greengenes + Vibrionales + Pseudoalteromonadaceae" search.

[b] These papers are also included in the tally for papers using Greengenes 13_5 or 13_8.

Although it is not clear how this incorrect assignment error in arose between Greengenes version 12_10 and 13_5, it does not seem to be due to a real or inferred dispute in taxonomic classification. We acknowledge that potential taxonomic errors are problematic for other databases, as well (*Edgar, 2018a*, *2018b*). Therefore, support for curation of taxonomic databases that are foundational to microbial ecology research using next-generation sequencing approaches is critical. This is an opportunity for field-wide collaboration to ensure that databases reviewed and curated.

## ACKNOWLEDGEMENTS

We thank M. Ghazaleh Bucher, J. Westrich, and E. Ottesen for comments and advice on this manuscript.

### Funding

This work was supported by NSF grant OCE-1357423 (to Erin K Lipp). The funders had no role in study design, data collection and analysis, decision to publish, or preparation of the manuscript.

### Grant Disclosures

The following grant information was disclosed by the authors:
NSF: OCE-1357423.

### Competing Interests

The authors declare that they have no competing interests.

### Author Contributions

- Keri Ann Lydon conceived and designed the experiments, performed the experiments, analyzed the data, prepared figures and/or tables, authored or reviewed drafts of the paper, approved the final draft.

- Erin K. Lipp conceived and designed the experiments, performed the experiments, analyzed the data, prepared figures and/or tables, authored or reviewed drafts of the paper, approved the final draft.

## Data Availability

The raw data have been provided as a Supplemental File.

## Supplemental Information

Supplemental information for this article can be found online at http://dx.doi.org/10.7717/peerj.5248#supplemental-information.

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
