# Peer review of "Taxonomic annotation errors incorrectly assign the family Pseudoalteromonadaceae to the order Vibrionales in Greengenes: implications for microbial community assessments"

_PeerJ, doi:10.7717/peerj.5248_

## Round 0.1 · original submission · Major Revisions

Dear Drs Lydon and Lipp,

I agree with all the reviewer comments and I think the paper can be accepted after you provide and answer all the reviewer suggestions.

Reviewer 1 ·

Basic reporting

no comment

Experimental design

no comment

Validity of the findings

no comment

Additional comments

The study is useful in its research area and suited for publication.

Reviewer 2 ·

Basic reporting

see below

Experimental design

see below

Validity of the findings

see below

Additional comments

In this paper, Lyndon and Lipp call the field’s attention to errors in recent versions of the Greengenes 16S rRNA gene taxonomic database. Sequences corresponding to Vibrio and Pseudoalteromonas are often misclassified using Greengenes, as evidenced by cross-checking with SILVA, RDP, and NCBI’s 16S database. The purpose to the paper is to provide evidence for the errors, raise awareness of the issue, and to call the field to better curate these databases.

The paper has two main results - (1) evidence that the errors are present in the database and (2) evidence that these errors have impacted the field boradly.

Result 1

The authors mention in the discussion there could be two types of errors - Vibrio that are misclassified as Pseudoalteromonas and Pseudoalteromonas that are misclassified as Vibrio. However, their data appears to only explore the latter problem.

I feel the strongest way to communicate result 1 would be a phylogenetic tree of the Alteromonadales and Vibrionales 16S sequences from the Greengenes database with indication on the tree the taxonomic classification of each sequence based on the different methods the authors tried.

In the methods section, it says BLAST, SeqMatch, and SINA were used but does not indicate how they were called. Was this done using web interfaces with default settings? A supplemental table explaining which parameters were chosen would aid in reproducibility.

Result 2

The authors used 86 papers to assess the impact. A table that lists each of these papers should be included.

Other comments

I think the introduction and discussion sections could spend more time explaining the ecological significance of Vibrio and Pseudoalteromonas and explain why correct classification could matter. Both genera are gamma proteobacteria, mostly aerobic, heterotrophic, conditionally rare taxa that often have the potential to eat a wide range of carbon sources. What are some broad ways they are different? In particular, I feel the first paragraph of the discussion doesn’t flow - the hypothesis of how the errors could have occured in Greengenes feels like a non-sequitur. However, the second paragraph is stronger and has a clear call to action.

Reviewer 3 ·

Basic reporting

The authors perform a comparison of annotation of Pseudoalteromonas by Greengenes Database and found an important problem in this Database. This study merits publication after a few corrections.

I enjoyed this short paper, but have a few remarks to improve the quality of the study.
1. The authors mention (Line 73) type strains, the most relevant material to compare with and to prove the possible annotation errors. However, the information is hidden in the supplemental material, a large excel table. In this table I could not find the the closest matches for the 164 Pseudoalteromonas sequences mentioned (Table 1). Which are the closest type strain for those 164 sequences? Which are the closest type strains for those 46 sequences of vibrionaceae (Table 1) ?
It is important to give more taxonomic information on those aspects and expand the taxonomic analyses using type strains. Table in supplemental material is not the way to present to the reader. It is not easy to read and interpret..
2. The taxonomic information will imply in ecological information that could be gained. Eg if Vibrio cholerae CT+ is present in the samples, one would argue for a possible risk for human health. If Vibrio coralliilyticus is present, one could argue for a possible risk for coral health. If many studies perform the wrong taxonomic affiliation they may came to wrong ecologic conclusions. Can the authors pinpoint a few example of such problems in the literature and expand the discussion?
3. Perhaps the authors could also include a phylogenetic analysis/tree with sequences from the Greengenes and from type strains. these will help to easier interpret the data and implications.

Experimental design

ok.

Validity of the findings

ok.

Additional comments

ok.

---

## Round 0.2 · accepted · Accept

Your article has been accepted for publication. Congratulations!

# Reviewer 2 ·

Basic reporting

The revisions are fine.

Experimental design

The revisions are fine.

Validity of the findings

The revisions are fine.

Additional comments

The revisions are fine. Thank you.